# Multispectral UAV Data and GPR Survey for Archeological Anomaly Detection Supporting 3D Reconstruction

**DOI:** 10.3390/s23052769

**Published:** 2023-03-02

**Authors:** Diego Ronchi, Marco Limongiello, Emanuel Demetrescu, Daniele Ferdani

**Affiliations:** 1CNR—Institute of Heritage Science, 00015 Rome, Italy; 2Department of Civil Engineering, University of Salerno, 84084 Fisciano, Italy

**Keywords:** landscape archaeology, 3D reconstructive modeling, Extended Matrix, photogrammetry, ground penetrating radar

## Abstract

Archeological prospection and 3D reconstruction are increasingly combined in large archeological projects that serve both site investigation and dissemination of results. This paper describes and validates a method for using multispectral imagery captured by unmanned aerial vehicles (UAVs), subsurface geophysical surveys, and stratigraphic excavations to evaluate the role of 3D semantic visualizations for the collected data. The information recorded by various methods will be experimentally reconciled using the Extended Matrix and other original open-source tools, keeping both the scientific processes that generated them and the derived data separate, transparent, and reproducible. This structured information makes immediately accessible the required variety of sources useful for interpretation and reconstructive hypotheses. The application of the methodology will use the first available data from a five-year multidisciplinary investigation project at Tres Tabernae, a Roman site near Rome, where numerous non-destructive technologies, as well as excavation campaigns, will be progressively deployed to explore the site and validate the approaches.

## 1. Introduction

In recent years, the integration of non-destructive technologies (NDT) for the identification, characterization, and investigation of hidden archeological sites has gained much importance due to the development of the performance and portability of sensors and the reduced cost of the technologies [1,2,3,4,5]. The growing number of examples integrating survey technologies into increasingly articulated processes has led to numerous attempts to use the data not only for investigation but also for simulation and dissemination of reconstructive hypotheses, at least partially transforming the survey results themselves into narrative elements useful for historical reconstruction [6]. In this paper, we first describe the combined use of multispectral photogrammetry (MPHG) and ground penetrating radar (GPR) for archeological surveys, and then experiment with three-dimensional modeling and visualization tools for data deriving from NDT. The combined use of multispectral imaging and GPR for archeological investigations has shown promise in some recent experiences [6,7,8,9]. Indeed, these two detection technologies are complementary: on the one hand, the flexibility of MPHG provides two-dimensional information about buried targets quickly and over a wide area [10,11], but often with a lower resolution, and on the other hand, the accuracy of GPR allows for detecting buried structures with detailed 2D/3D geometries, but with a longer acquisition and processing time, often limiting the area of investigation [12]. The subsequent use of 3D modeling software again had a dual purpose: (i) to test whether these tools can improve the visualization and interpretation of GPR data by integrating other sources (orthophotos, DEM, etc.) into a single platform; (ii) to apply the formal language Extended Matrix (EM) to make the process of GPR data interpretation both traceable and consultable. However, the use of heterogeneous data for 3D reconstruction requires a clear, transparent, and reproducible methodology according to Open Science specifications to verify data quality, reliability, and the scientific activity behind data processing and production [13]. The scientific debate on virtual archeology, a discipline that emerged in the 1990s along with the development and diffusion of digital technologies [14], gradually shifted to the discussion of approaches to reconstructive modeling and its mediation to ensure methodological rigor for the application of digital visualization methods. The London Charter (LC) of 2008 [15], which further developed the premises formulated in 2003 in the UNESCO Charter on Digital Heritage Conservation [16], regulated the principles of visualization in the field of cultural heritage. The conditions for the applicability of LC, especially in relation to archeological heritage, were improved in 2009 with the implementation of the principles of the Seville Charter (PSC) [17].

Only a few recent experiences with the combination of NDT and reconstructive modeling can be found in the literature: the Roman city of *Ammaia*, the Roman villa of *Durres*, and the Roman forum of *Carnutum.* In the first case, the ancient road networks identified by NDT investigation methods were interpreted using 3D reconstructions and visualizations [18]. In the second case, aerial photography and GPR surveys were combined to locate a late Roman villa. The data integration allowed hypothesizing the appearance of the villa and proposing a reconstructive model that also takes into account archeological and historical literature. In addition, the reconstructive models provided different levels of confidence to inform about reconstructive decisions, but without providing explicit criteria for formalizing them [19]. Finally, a new integrated approach for 3D GPR interpretation has recently been presented that focuses on the visualization of heterogeneous data and supports the joint visualization of 3D scenarios composed of GPR volumes, 2D prospection images, and three-dimensional reconstructive models [20].

As mentioned earlier, there are few and relatively recent cases in the literature where 3D modeling techniques have been integrated with geophysical and geomatic data in archeological contexts. In most cases, the goal is to improve the visualization of geophysical data to enable a more or less schematic virtual reconstruction of buried architectural remains [6,21,22,23]. Apart from the above examples, the visualization of the results of GPR surveys, both three- and two-dimensional, is usually carried out using proprietary software developed with a monothematic approach to the representation of buried targets and whose algorithms are severely limited in terms of customization and the type of information they can manage [24]. To overcome this condition and open a new research direction, this paper proposes a data management process in Blender (an open-source software) using a set of original add-ons (Python-based) that will be progressively improved and hopefully reused by other research groups (GPL3 license), so that data management is conducted in a complete computer graphics environment for 3D polygonal modeling. In addition, it is proposed to use open formats to achieve better integration of data, often coming from different disciplines (photogrammetry, multispectral data, source-based 3D models, etc.). It is also noted that the general tendency is limited to the representation of 3D data without semantic formalizations and an explicit link to a data source database, the latter being particularly relevant for 3D reconstruction due to the heterogeneity of sources [25]. To address this current state, we present a first application called Extended Matrix geophysical Workflow (EMW-geo v.0). Starting with version 1.3, the EM language was developed to provide specific common workflows for different data to establish a “common ground” between disciplines. Therefore, one of the goals of this work was to provide the basics of EM for a geophysical workflow: first for anomaly acquisition and publication, and second as a source for interpretation and three-dimensional reconstruction. In this case, 3D modeling is used to improve data perception and management through “augmented perception” within a single 3D platform to design reconstructive volumes for buried and invisible structures. The 3D reconstruction is, therefore, limited to the visualization of shapes, as a more realistic representation based on GPR data alone is not possible. Schematically, the objectives of this article can be presented as follows:Validation of a protocol for the combined use of sensors in archeological surveys to support 3D reconstruction at the territorial scale. This protocol is formulated according to the logic that a lower intensity corresponds to a greater extent of investigation and vice versa [26,27]. This means that intensity is a combination of acquisition and processing time, implementation cost, lower impact on remains, and greater effectiveness of NDT compared to archeological excavation alone [28,29].Ground truthing for identified GPR anomalies using archeological excavations.Development of a workflow for managing GPR data (time slices) using Blender software (Blender: 3D Survey Collection, new version 1.6) by developing original tools for visually filtering and cleaning the data.Formalize geophysical EMW (with addition of extended matrix layer) to track process transparency, read and interpret anomalies, and create 3D reconstructions using GPR data.

The next section deals with the material (the case study) and the methods used (tools and pipelines grouped into integrated methodological workflows). Section 3 presents the results of the work with special attention to the derived data and their interpretation. Section 4 is dedicated to the discussion of the results in a broader perspective, including the limitations of the proposed method and the future perspective.

## 2. Materials and Methods

In this paper, a three-step method is described, namely: (i) detection, (ii) ground truth verification, (iii) 3D reconstructive modeling. After selecting a case study (Section 2.1) that offered the best opportunity to test the proposed method, the first activity was data collection. The decision to use a multispectral sensor to detect cropmark anomalies in a 55 h area was based both on its accessibility in terms of size and economy and on the relative speed of data acquisition and ease of processing (Section 2.2). These critical elements have led to the widespread use of this method in archeology in recent years, particularly in the identification and characterization of archeological structures buried in cultivated fields [30,31]. Once the anomalies were identified, their nature and consistency were further investigated using a GPR (Section 2.3). This geophysical survey tool, which is widely used in archeology [32,33], can be used to investigate elements in the subsurface. This sensor was chosen because, in addition to the geometric accuracy that generally characterizes the data generated, it offers the possibility of generating “time slices”, i.e., plans of the subsurface at different depths. Following the outlined scheme (i.e., decreasing the surveyed area with increasing survey intensity), after the NDT implementation phase but before the interpretive and reconstructive activities, ground truth verification was assessed by conducting some archeological excavations in accordance with the GPR anomalies (Section 2.4). After the ground truth verification, also through the development of specific software add-ons (Section 2.4.1), the work continued in a computer graphics environment to produce a volumetric reconstruction in accordance with the anomalies and the excavation data. In summary, the objectives described above (Section 1) were pursued through the sequential implementation of the following techniques and procedures (see also graphic flowchart in Figure 1): Selection of an archeological landscape suitable for testing the proposed methodology (Section 2.1);Multispectral photogrammetric survey (MPHG) (Section 2.2);Multispectral map processing and anomaly identification (Section 3.1);Photogrammetric RGB survey (PHG) for the selected area before in-depth analysis with GPR technique (Section 2.2);GPR acquisitions (Section 2.3), time-slices processing, and anomalies identification (Section 3.2);Verification of ground truth in accordance with GPR anomalies through archeological excavations (Section 2.4);RGB photogrammetric survey (PHG) after excavations (Section 2.2);Management and visualization of DEM, orthophotos, and time slices in Blender (Section 3.3);Anomalies representation using 3D proxies and data interpolation (Section 3.3);Implementation of the extended matrix (EM) to support traceability of the information used in 3D modeling (Section 3.3).

### 2.1. Case Study’s Definition and Relevance: Tres Tabernae

The site under study is a Roman *mansio*, i.e., a resting place on a main road where one could stay overnight, eat, and change horses. *Tres Tabernae* [34] was, until the recent excavations, only a toponym mentioned in literary sources and historical maps, located between the *mansiones* of *Ad Sponsas and Forum Appii*, in the section between the XXV and the XLIII mile of the Via Appia. The excavations carried out in 1993 by the Soprintendenza per i Beni Archeologici del Lazio (SABAP) made it possible to locate the site in the locality of “Piscina di Zaino”, on the edge of the modern Appian Way, as shown in Figure 2. Although the buildings must have been originally distributed evenly along this section of the Via Appia, today, only a small part of this complex is visible, as shown by previous geophysical and archeological surveys [35]. Under the present visibility conditions, influenced by the opening of several extensive but isolated excavation areas and by the very modest preservation of the structures due to agricultural works, it is difficult to analyze the functional distribution of the spaces, the relative chronology between the isolated areas, and the actual extent of the complex. For practical clarity and for documentation purposes, the visible areas were numbered, as shown in Figure 3. The first visible area (AREA I) is the one located immediately before the entrance of the fenced area. It is an area composed of small rooms that use a narrow-paved road as an axis of symmetry and have an eccentric orientation with respect to the present Appia and the rest of the structures. From this street, one could enter a driveway that has several phases of life. This space, in turn, led to a series of others distributed on this side of the street. Other structures were located on the opposite side of the street. The space between these rooms and the Via Appia is occupied by a cistern that was partially demolished for the construction of a canal. In the area of this excavation front, towards the other excavated areas, part of a small bath complex is currently visible. About 70 m east of the road (AREA I) is another excavated area (AREA II), brought to light by SABAP in the 1990s. The area consists of a series of rooms using a corridor as an axis of symmetry, with an eccentric orientation both in relation to the course of the present Via Appia and in relation to the structures visible in areas I and III. To the west, about 10 meters from area II, there is another area that is the subject of excavations (AREA III). In this case, the state of conservation of the elevations is very poor due to agricultural activities.

This area, which seems to follow a *domus*-like division of space, is organized around an *atrium* with an *impluvium*. Around this central space, accessible from the Via Appia through corridors divided into three rooms, cubicula of modest dimensions unfold. Immediately behind the atrium is a space that we could very cautiously interpret as a tablinum, assuming for this part of the *mansio* an arrangement similar to that of a *domus*. The use of Tres Tabernae to illustrate the discussed methodology was not accidental, it was guided by a number of considerations, both internal and external to the research project. First, it should be noted that the area to be subjected to intensive investigation (i.e., GPR and archeological excavation) was not one of the areas identified by the MPHG in the first place. This aspect, which could limit the experimental value of the presented procedure, was determined by the legal and organizational impossibility of conducting archeological excavations on private property and, at the same time, by the availability of a site, precisely the selected one, located on a public property currently involved in a study and restoration project. As described in the following section, the complexity, extension, and historical stratification of the site, as well as the contribution of the available heterogeneous sources (historical, archeological, etc.), also make the case study particularly suitable for the construction of a semantic model using EM formal language [36,37,38,39]. From a technical point of view, it should be noted that although the area was selected according to external criteria for MPHG identification as well as presenting excellent conditions for conducting GPR surveys ( flat, clear area with low thickness of the soil and strong contrast), it is part of a settlement network structured around secondary roads that, on the one hand, will be further investigated in the next four years with the same protocol and, on the other hand, are very fruitful for the analysis of road trajectories.

### 2.2. Photogrammetry in Archaeological Prospecting: Background and Methodological Overview

In the field of photogrammetric technology for archeological prospection, multispectral sensors are increasingly used in addition to RGB ones. These sensors are capable of detecting the amount of energy reflected from objects on the earth’s surface in the different wavelengths of the electromagnetic spectrum (generally near-infrared and red-edge). Photogrammetry is based on the acquisition of images processed with specific software that uses the principles of multi-image photogrammetry using Structure from Motion (SfM) algorithms [40,41]. These methods allow the restitution of dense point clouds, 3D models, DSM (digital surface model), and orthomosaics with high final accuracy [42].

RGB sensors are used primarily when the structures to be detected are in an area with good or excellent visibility, or when there is no vegetation layer such as trees or shrubs [43,44]. Over the past decade, sensors have become increasingly miniaturized, allowing compact and lightweight cameras to be incorporated into UAVs, favoring their development and performance. Unlike an RGB sensor, a multispectral one provides a multiband image that, by analyzing the spectral response within the different bands captured, allows classifiers to extract territorial information and create accurate thematic maps. For example, with a multispectral camera positioned on a UAV, it is possible to detect differences between areas with different vegetation vigor and create vegetation indices defined as mathematical reflectance combinations of vegetation in different spectral bands [11].

#### 2.2.1. Multispectral Photogrammetry in Archaeology: Background and Methodological Overview

The identification of positive and negative cropmarks using vegetation indices derived from UAV or satellite imagery is almost exclusively discussed in the literature using a multispectral approach [30]. Indeed, most combinations of bands in multispectral imaging provide some information about the state of the plant, which, in the case of cropmarking, is related to the presence of anthropic elements covered by the soil (Figure 4). The spectral bands important for crop monitoring are green (555–580 nm), red (665–700 nm), *NIR* (740–900 nm), and red-edge (725–745 nm). By combining the pixel intensities of at least two bands, it is possible to create vegetation indices in the desired band. As mentioned earlier, these indices primarily use three spectral bands (green, red, and *NIR*) that are related to spatial differences in the density and physical condition of vegetation and soil moisture and thus document indications of the presence of buried archeological remains [45]. Various UAV systems can be used to collect territorial scale information to create high-resolution vegetation indices. Generally, a fixed-wing or multirotor system is chosen depending on the extent to be surveyed, the GSD (ground sample distance) required, and the sensor available [46]. In this use case, the aerial photogrammetric survey—with a multispectral sensor—was performed using a commercial fixed-wing UAV system, a Sensefly eBee X-AgEagle company, Cheseaux-sur-Lausanne, Switzerland—(Figure 4). This UAV has a central body made of carbon fiber, while the detachable wings are made of polystyrene and a tail propeller provides propulsion. In the central part, there is a high-performance IMU (inertial measurement unit) module and a GNSS L1/L2 module that can determine the centimeter position of the drone in real time by using correction data provided by a base station on the ground or by a permanent station network. The hardware weighs about 1.2 kg without the camera installation. In automatic flight mode—via a flight plan programmed in the dedicated eMotion application—two GNSS modes are available for both flight and georeferencing of the images. For our acquisitions, RTK/PPK correction mode was set prior to flight in order to post-process the data. Specifically, GPS /GNSS observations acquired from a ground-placed Geomax Zenith 20 Pro GNSS receiver–Geomax company, Ancona, Italy—were used to re-evaluate the flight path and record accurate GEOTAGs for the images. To further improve survey accuracy, eleven GCP (ground control points) marked by a 60 × 60 cm target were surveyed. The survey took place on 15 April 2022; 3928 images were acquired with a Parrot Sequoia–Parrot company, Parigi, France—multispectral camera, 982 for each band (Figure 4). This sensor is capable of capturing images in the visible and *NIR* spectra and provides calibrated data for optimal monitoring of vegetation health and vigor. The sensor has five integrated cameras that can capture green (~550 nm), red (~660 nm), *NIR* (~790 nm), and red-edge (~735 nm). The four multispectral bands are captured by a 1/3.2" sensor (1280 × 960 pixels, pixel size 3.75 µm, focal length 4 mm). A 16 MPixel RGB sensor is installed on board as the fifth camera (not activated on this flight). It measures 6.17 × 4.55 mm (4608 × 3456 pixels, pixel size 1.34 µm) and has a focal length of 5 mm. The system also has a fully integrated, sky-facing sunlight sensor that detects and records current lighting conditions and automatically calibrates the camera outputs. Parrot Sequoia was radiometrically calibrated prior to flight using the dual-sensor pair calibration method proposed by Jin and Eklundh [47]. The method involves radiometric calibration of sensors by using the sun as an illumination source. Radiometric calibration is performed by having an upward-facing sensor register incident radiation and a downward-facing sensor detect a reflectance value using an appropriate calibration target. The flight lines were designed using the Emotion version 3.13.0 Build 236 software package. For this survey, an altitude of 65 m above the take-off point was set as the target. This software package also estimates flight altitude using elevation data derived from Google Earth to check for overlap with elements that are part of the overflown area. The parallel flight lines were programmed to overlap 80% to the front and 60% to the side. The flight took approximately 18 minutes to cover the planned area of 55 hectares. Image acquisition programming took into account the requirements of the project—a ground sampling distance (GSD) of about 6.0 cm/px. Image processing was performed using Pix4d Mapper software version 4.5.6, a dedicated suite for UAV image processing that has several preset templates optimized for resolution and time depending on input data and desired end result. The Pix4Dmapper software version 4.5.6 has dedicated processing models for Sequoia cameras; in particular, it is possible to use the Ag Multispectral module for *NIR*, red-edge, red, and green. The software uses a classic SfM pipeline that can be summarized in the following steps: (1) computation of internal and external orientation parameters (with automatic detection of key points and tie points) and creation of a sparse cloud, (2) extraction of a dense cloud, (3) construction of a polygonal model, (4) creation of a dense cloud of a polygonal model, (4) texture mapping, and (5) creation of orthophotos and possibly vegetation indices [48]. As mentioned earlier, image processing in Pix4D was performed using full-scale images, with image pair matching set to Aerial Grid or corridor. The target number of key points was set to 20,000. An image reduction factor of 1/2 per side (1/16 of the total resolution) was used to create the dense point cloud. For this project, the estimated value of RMSEE and RMSEN for the GCPs was about 4 cm, while the RMSEh was about 9 cm. After processing the orthophotos for the four bands using the “Index calculator” tool in Pix4DMapper, it was possible to calculate and export the raster with a pixel size of 10 cm in the *NIR* and red-edge bands, as well as the multispectral vegetation indexes *NDVI* and *GNDVI*.

#### 2.2.2. RGB Photogrammetry 

The implementation of RGB photogrammetry does not offer any innovative aspects compared to what has already been described in detail in the literature. Therefore, we include this paragraph only to give a complete description of all the tools and procedures used. In order to obtain the required documentation (orthophotos and DEM) for both the planning of the GPR investigations and the ground-truthing verification, two photogrammetric flights were performed, one before and one after the excavation. A DJI Mavic 2–DJI company, Shenzhen, China—(Figure 4) equipped with a 4/3 sensor with a resolution of 20 Mpx was used for both acquisitions. Both flights were conducted using the same flight plan, which allowed the acquisition of 275 pseudo-nadir images at an average altitude of 35 m. The data were processed using Agisoft Metashape 2.0 [49], aiming for a GSD of 1.6 cm for both orthophotos and DEM. Prior to flights, six GCP with targets (60 × 60 cm) were surveyed on the ground using Geomax Zenith 20 which were later used for scaling and error estimation (RMSE and RMSEN 1.2 cm).

### 2.3. GPR Survey

Target identification with MPHG was followed by GPR prospecting to further investigate the anomalies in terms of type and consistency. GPR is a widely used tool for surveying ground discontinuities and is frequently used in archeology [50]. The processing of GPR data is mainly done with specialized software that is rarely open-source, at least for now [51,52]. These tools allow, on the one hand, the processing of raw signals and, on the other hand, the generation of three- and two-dimensional deliverables. For the GPR acquisition, as shown in Figure 5, an IDS HiMod was used, consisting of a single channel dual frequency 200/600 MHz antenna, a Radar control unit DAD (digital antenna driver), and a laptop (Panasonic Tough-Book CF -19) equipped with the K2Fastwave 2.0 acquisition software responsible for system management and data storage. After signal calibration tests, the following instrument configuration was chosen: full scale times 60 ns for the 600 MHz antenna and 120 ns for the 200 MHz antenna, 16-bit dynamic range, 512 samples per trace. This configuration, as shown in Figure 5, allowed investigations to a depth of about 2.0 meters with the 600 MHz antenna (Figure 6) and up to 3.5 meters with the 200 MHz antenna (Figure 5b). Prior to the GPR survey, thirty 25 × 25 m square-shaped acquisition areas were traced on the ground using a GNSS Geomax Zenith20 (Figure 7a). The areas were marked with plastic stakes and delineated with nylon cords. GPR measurements began immediately after this preliminary work. The campaign was conducted in line scan mode, with the antenna pulled along the profile in continuous recording, which allowed the acquisition of equidistant longitudinal profiles every 50 cm. Therefore, as shown in Figure 7, 547 parallel profiles of varying lengths were recorded during the surveys, covering a total area of approximately 1.8 hectares. A local Cartesian reference system was established within each acquisition square, using the sides of the quadrat as the X and Y axes. A taut tape was used to maintain the spacing between the profiles, while an odometer was used to determine the positioning of the antenna along the profiles. To ensure better positioning quality, the vertices of the squares were surveyed again with the GNSS system after the GPR surveys. After data acquisition, the data were transferred from the memory of the acquisition unit to a desktop PC, where they were processed using the GPR-Slice v7 software [53]. The following filters were applied to the recorded signals: bandpass filter [54,55], background removal [56], and migration [57].

### 2.4. Validation through Archaeological Excavation 

As shown in Figure 8, to verify the reliability and accuracy of the GPR data, seventeen stratigraphic excavations were conducted prior to 3D reconstruction, covering a total area of 350 m^2^ (about 2% of the area’s extent). The excavation areas were selected according to the anomalies identified in the time slices and positioned on the area using GNSS instruments. As can be seen in Figure 9 and Figure 10, some examples of the excavated archeological structures show a good overall correlation with the isosurfaces, with average vertical differences of +/− 20 cm.

#### 2.4.1. Extended Matrix: NDT Data Management and Visualization in Blender

The project of 3D simulation at Tres Tabernae is based on a theoretical and methodological approach related to the principles of “transparent integration” of data. As could be expected, the approach developed within Blender software was used by a special open-source add-on: EMtools for Blender [58]. With this approach, it was possible to record, publish, and interpret GPR anomalies and use them as a source for reconstructive hypotheses. The interpretation process always requires a certain degree of caution because of the risk of creating models based on suggestions or superficial impressions that do not take into account the different data sources (excavations, GPR, multispectral data, etc.).

From a practical point of view, the implementation of EM is a two-step workflow: (i) collecting and reading data, and (ii) adding EM graphs (in our case, using the yED software version 3.22) that map the collected information along with the details of the interpretation process (i.e., sources, interpretations, etc.). To enable this connection and add data from geophysical surveys, a mapping table between GPR anomalies and the EM language (extended matrix workflow (EMW) Geophysics ver. 0) is presented, hopefully making this method reusable and implementable (Table 1).

Visualization and management of data using 3D modeling tools enabled better understanding and management of complex and heterogeneous datasets, improving analysis processes [59]. In turn, the implementation of EM guarantees the intellectual and technical honesty of the interpretative work, making it “transparent”. EM is a formal language that emerged from the evolution of a tool already used in archeology for stratigraphic documentation: the Harris Matrix [60]. The term “extended” indicates that the matrix includes and defines not only the actual archeological stratifications, but also their hypothetical reconstructions, the so-called virtual stratigraphic units (VSU) [37,40,61]. So far, EM has been used mainly for virtual reconstruction projects of excavated archeological sites [62,63] and for the simulation of ancient architectures based on evidence (preserved or collapsed structures found during excavations) or other historical and archeological sources (e.g., archives, iconography, comparisons, etc.). In this case study, we present an initial version of the EMW-geo (geophysical workflow) in which GPR anomalies are represented as documentary stratigraphic units (USD), meaning that they are investigated in a “deferred” mode, i.e., a photograph of a wall of a monument that no longer exists. This situation is similar to that of GPR sensors, where structures cannot be seen directly, but can be guessed from a “map” of anomalies and then traced in an interpretive way to the structures that produce them. Thus, thanks to EM, it is possible both to formalize the interpretation of the anomalies (USD) and to integrate them from a planimetric/volumetric point of view with the USV/s (structural virtual stratigraphic unit), which represents the “geometric restoration” of what is considered objective. Both categories of SU are enriched with metadata (geometric, geophysical, multispectral measurements; historical, archeological, and architectural sources, etc.) and paradata (the documentation of logical-interpretive processes that led from archeological evidence to interpretive hypotheses). This approach, previously used only in archeology, is here declined to manage GPR data like any other historical or archeological source, modeling the “unseen” in three dimensions and supporting reconstructive interpretation. The analysis and interpretation of large data sets in archeology (i.e., archeological, geophysical, geomatic, etc.) is a complex task that may require the integration of multiple sources. The visualization of such data in a 3D semantic system based on EM is a fundamental step to facilitate and improve this process. EM, since version 1.2, has a general workflow (EMW) structured in five steps from data acquisition to presentation [37]:Data collection: collection of sources (geomatics, geophysics, archeology, history, etc.);Data management and analysis: design of proxy models for existing structures semantically enriched with stratigraphic information;Implementation and virtual reconstruction: design of proxy models for reconstructive hypotheses based on archeological evidence and sources. The proxy model is then linked to the EM and assigned different color codes representing different levels of confidence.Representation model: design of a textured and shaded reconstructive photorealistic mesh;Publication and dissemination: rendering of the mesh model according to the chosen esthetic style and publication context.

One of the goals of this paper, as mentioned earlier, was to specialize the EM workflow to include geophysical survey results as a standard source. Since this is a preliminary application limited to geophysical data from GPR and archeological features (formalized in general EMW), we decided to limit the experiments to the first three steps. All detailed 3D and 2D data listed below were imported into Blender software in geographic coordinates (EPSG:32633). For this purpose, we used a combination of the open-source add-ons 3DSC [64] and BlenderGIS version 2.28 (Delaunay Voronoi and Triangulation Diagram, Blender GIS, domlysz, https://github.com/domlysz/BlenderGIS, accessed on 25 February 2023), which are designed to integrate georeferenced territorial data while meeting computer graphics requirements.

Terrain and visible structures 3D model;Time slices;Multispectral maps;RGB orthophotos.

The time slices were further processed in Blender by mapping them to meshes and applying the “shrinkwrap” modifier, which allows a mesh (time slice) to fit its topology to the surface of another object (DEM) by moving each vertex as close as possible to the reference surface. In this case, the projection was limited to the Z axis, where vertices could be shifted in both positive and negative directions, and an offset value equal to the depth of the individual slice was associated. Each mesh (slice) was associated with another subsurface modifier with a value of 10 subdivisions to increase the accuracy step of the draping of the geophysical data on the ground (about 19.7 cm on the x and y axes between the different overlapping points obtained with the shrinkwrap between the slice and the ground). In this way, it was possible to position all GPR time slices at the correct height with respect to the ground surface. Subsequently, the time slices/meshes were filtered using the “Raster Filtering” algorithm in 3DSC (formalized by the “Material” nodes in Blender, Figure 11) to highlight anomalies at different depths. This allowed a better visualization of the data and a better understanding of their position in relation to the visible structures and the structures identified during the verification tests.

The anomalies highlighted on the time slices are an incomplete representation of what is underground. Some elements “respond” differently to the sensor depending on local ground conditions, while in other cases, the anomalies are difficult to read due to the effective fragmentation of the buried elements. The operator interpreting the anomalies traditionally maintains an overview of the alignments, shapes, and thicknesses, drawing on personal experience and possibly comparisons confirmed by excavations. For data interpretation, each time slice is equivalent to a different document that may be located at a different depth beneath the overburden. These documents must be read together to extrapolate the characteristics of the object that represents the anomaly. The work of interpretation in a three-dimensional environment made it possible to design volumes, called proxies, that represent a synthesis for different sources and, in particular, in this case, for time slices and discoveries made with archaeological excavations. These elements were represented within the EM graph in the form of "nodes" or stratigraphic units (actions performed by humans or nature that led to the creation of a tangible unit: a wall, a pit, a floor, a pillar, etc.). After highlighting visible elements by the anomaly map (USD, in orange), these elements were recovered by integrating the gaps between them (USV/s, in blue). Figure 12 shows a preliminary relative stratigraphic sequence between units based on the depth determined by the GPR.

## 3. Results and Discussion

### 3.1. Multispectral Maps

Vegetation discontinuities, known as cropmarks, are phenomena characterized by seasonal visibility and are often associated with agricultural rhythms. This type of anomaly can be positive or negative depending on the phenomenon that influences its formation. This distinction between positive and negative cropmarks is related to the vitality of vegetation, which depends on the richness of the substrate on which plants can feed. For example, the presence of filled ditches leads to taller and more abundant plants (positive cropmarks), while the presence of walls with low nutrient layers leads to a vegetative crisis (negative cropmarks). Cropmarks are among the most important proxy indicators for buried archeological remains, and their characteristics depend on land use, meteorological parameters, soil type, and vegetation. The MPHG implementation therefore aimed to test the reliability and productivity of airborne multispectral systems as the first cost-effective tool for detecting cropmarks of archeological interest and, if possible, alignments between them. Thus, this approach aimed to narrow down the areas to be investigated with other techniques of greater intensity (i.e., magnetometry, ground penetrating radar, and archeological excavations). The main advantage of this approach is that the use of UAVs provides very detailed results—on the order of cm/px—covering large land areas in a short time and with manageable processing times. The weakness of this technique in archeological prospection is its susceptibility to vegetative growth of plants, and thus the limited predictability of results, and the difficulty of using it—except in particularly favorable cases—within identified buildings to study internal layout and spatial relationships. The following spectral maps and vegetation indices were calculated from the information obtained with the multispectral sensor (Figure 13):*NIR*: Near infrared is one of the most productive spectra for cropmark detection [65,66]. More specifically, green and healthy plants tend to show high values in the *NIR* band, while stressed vegetation is characterized by a low *NIR* value due to water or nutrient deficiency. This fact makes the *NIR* band ideal for identifying archeological cropmarks related to changes in the growth and/or color of vegetation compared to the surrounding environment. In the *NIR* band (700/750 nm to 1400 nm), the spectral characteristics of leaves are no longer determined by pigments. The very steep increase in energy reflectance between 700 nm and 750 nm is located exactly in the “red edge” zone, i.e., the most important feature in the reflectance spectrum of healthy vegetation. This transition zone with a very abrupt change in reflectance results from the fact that absorption by pigments is low and *NIR* reflectance increases, leading to one of the most extreme slopes in the reflectance spectra of natural materials. It should also be noted that the red-edge band is a sensitive spectral band that helps to improve the accuracy of plant classification and, thus, the identification of anomalies [67]. The following indices were calculated using the multispectral camera:*NDVI*: Normalized difference vegetation index is a measure of healthy green vegetation [46,68]. *NDVI* was calculated using the following equation from near-infrared (790 nm) and red (660 nm) reflectance measurements (with 40 nm full-width half-full-maximum bandwidth) of the spectrum and band values from −1.0 to 1.0.
(1)$$NDVI=\frac{R_{NIR\;}-R_{RED\;}}{R_{NIR\;}+R_{RED\;}}$$


*GNDVI*: Green normalized difference vegetation index is similar to *NDVI* except that it measures the green spectrum from 540 to 570 nm instead of the red spectrum [30]. It is an index for photosynthetic activity. It is a chlorophyll index and is used at later stages of development because it saturates later than *NDVI*. It is one of the most commonly used vegetation indices for determining water and nitrogen uptake in the plant canopy.


(2)$$GNDVI=\frac{R_{NIR\;}-R_{GREEN\;}}{R_{NIR\;}+R_{GREEN}}$$where, for each equation, *R_RED_*, *R_GREEN_*, *R_NIR_*, are, respectively, the red, green, and *NIR* bands and the reflectance value at x nanometers.

### 3.2. GPR Time Slices

As said, even if the next version of EMW-geo implements more data types from NDT, in the current stage of development, as also presented in the recent literature [53], this work focuses exclusively on the use of time slices, i.e., subsoil images processed for different depth intervals. Consequently, the processing and display technique known as time slices [69] was implemented to overcome the drawback associated with the analysis of individual profiles and to obtain more readable results. This technique consists of interpolating all the acquired profiles to obtain a planimetric image of the subsurface at a constant depth. In the case presented, the time slices were computed at intervals of 3 ns for the data acquired with the 600 MHz antenna and at intervals of 6 ns for the data acquired with the 200 MHz antenna. A total of 40 time slices were processed for each of the frequencies used. Only a selection is shown in Figure 14 and Figure 15. The depth of the anomalies is expressed as the delay between the transmitted and reflected pulse. The value of the delay is converted to a metric value when the propagation velocity of the signal in the medium under study is known. The propagation speed was calculated using the hyperbelfit method, estimating an 0.080 m/ns speed.

### 3.3. Transparent Integrated Dataset and Standardized Views of the Tres Tabenae Site

The results obtained with the presented method, i.e., the 3D bird’s-eye views, offer a clear contribution to the visual understanding of data sets with multiple sources. In particular, for the photogrammetric data (textured mesh), the problem arose of how to show the information “hidden” under the surface. Indeed, the photogrammetric survey represents the current state of the ground, while the GPR anomalies are located at known distances below this level. After experimenting with the visual metaphors of surface transparency, which make the underlying information barely legible, we decided to split the terrain highlighting the volumes and the underlying anomalies (Figure 16). These anomalies, positioned at their correct elevation above sea level, were filtered according to the methodology described earlier so that interpretive USV/s proxies could be drawn. These proxies then assumed not only the planimetric XY position, but also the elevation, which is in a range between a minimum and maximum given by the anomalies and the ground elevation (Figure 17).

Proxies were auto-color-coded according to the EM formal language, using red for ground truth (USM), orange for GPR anomalies (USD), and blue for integration (USV/s).

Information associated with proxies took into account the possible difference in the anomaly’s chronology as visible from both ground truth and GPR anomalies orientation; the add-on automatically converted such difference in a color code, as shown in Figure 16. Volumetric 3D representation, as a simplified version of the virtual nonstructural stratigraphic units (USV/n), represents a novelty in the EM workflow specifically formalized for modeling based on NDT data.

## 4. Conclusions

This paper presents a three-step method, namely: investigation by combining MPHG and GPR, ground truth verification with archeological excavations, data management, and 3D reconstruction in Blender. The presented approach, related to the investigation phase, offered remarkable results both from a procedural point of view and in terms of cost/time ratio. The implementation of MPHG allowed, apart from what was previously discussed (Section 2.2.) regarding the selection of the case study, to reduce by about 50% the areas that had to be subjected to a GPR investigation. In the same way, the GPR reduced the amount of archeological excavation that would have been necessary to planimetrically survey the entire area by about 90%, eliminating the chance factor of excavations based only on direct observation of the area. Clearly, stratigraphic excavation offers significant advantages over GPR prospecting in terms of information obtained (i.e., relative and absolute chronology of structures, typology, and direct observation of construction techniques and coatings). However, when it is not possible to conduct extensive excavations or plan in advance for temporally distributed interventions, the illustrated approach has undeniable advantages, particularly in terms of excavation planning and understanding of the entire site. Excavation of a limited area, as documented by the illustrated experience, combined with the use of GPR, in turn, allows us to partially address the lack of chronological information associated with the exclusive use of NDT for site investigation. The management and 3D modeling of NDT data in Blender were integrated into the interpretation process by supporting it with simulations. Indeed, the management of 2D and 3D multisource data (multispectral analysis, GPR, archeological survey) within a 3D semantic platform has enabled a better interpretive capability. The presented method also allowed: (i) modeling interpretive volumes by comparing in real time with different GPR layers and emerging structures; (ii) interrogating in real time the nature and depth of anomalies to conjecture different alignments based on depth; (iii) interpolating volumes to reconstruct, when possible, identified structures. Moreover, the interpretative act was always transparent and was represented with a standardized color palette (following the indications of the Open Science project Extended Matrix) that allowed us to distinguish between the objective data (surface structures) and the interpreted data (interpretation and interpolation of anomalies). This three-dimensional, multi-layered data management thus ensured a higher level of operational flexibility by using 3D modeling tools not only for visualization, but also as part of the interpretation process. To achieve this higher level of understanding, processing, and communication, the first version of EMWgeo v.0 was presented in a standardized workflow for geophysical data within the Extended Matrix. The experience gained so far also allowed us to focus on aspects that need further testing and development. From a procedural standpoint, validation of the procedure would require implementation in a newly identified target area while investigating a less favorable area. From a technical perspective, given the highlighted difficulties in using MPHG data for 3D reconstruction, the magnetometric technique will be used in the next campaigns. This decision results firstly from the attempt to further reduce the extent of the target area and secondly to contribute with more meaningful data to the 3D reconstruction on a territorial scale. For future development of EMW-geo in conjunction with time slices, it is conceivable that GPR point clouds could be used as objective input data to be used in algorithmic filtering processes for semi-automatic modeling of gaps in proxies (USD).

## Figures and Tables

**Figure 1 sensors-23-02769-f001:**
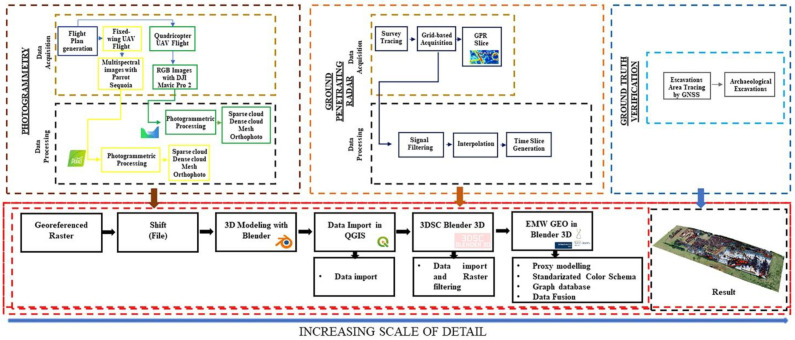
Graphic flowchart.

**Figure 2 sensors-23-02769-f002:**
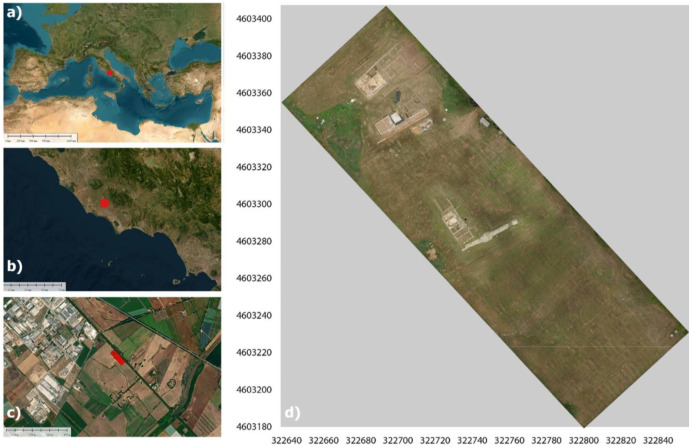
Test area: (**a**,**b**) location map; (**c**) the area of the “Tres Tabernae” (Italy); (**d**) UAV orthophoto of the visible archaeological remains.

**Figure 3 sensors-23-02769-f003:**
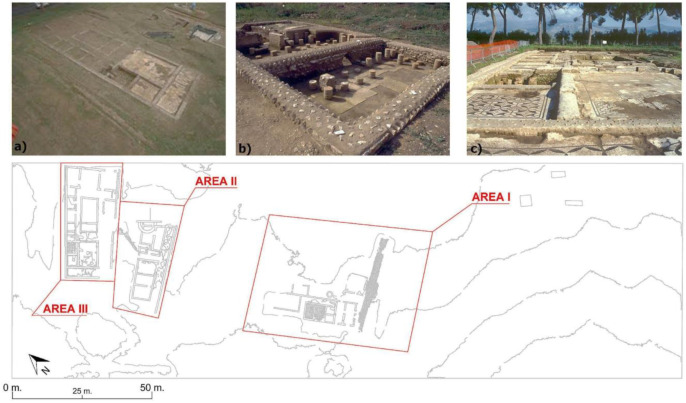
Visible buildings before ground truth verification: (**a**) UAV image of Area III; (**b**) bath complex; (**c**) structures in Area III paved in mosaics.

**Figure 4 sensors-23-02769-f004:**
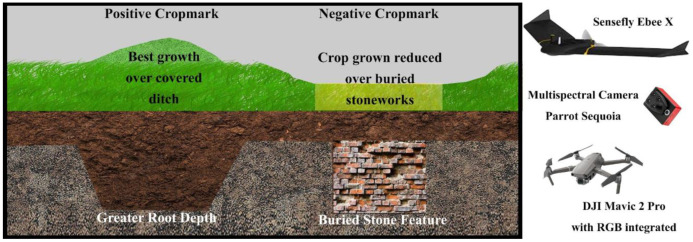
Positive and negative cropmarks with instruments for photogrammetric surveys from UAVs.

**Figure 5 sensors-23-02769-f005:**
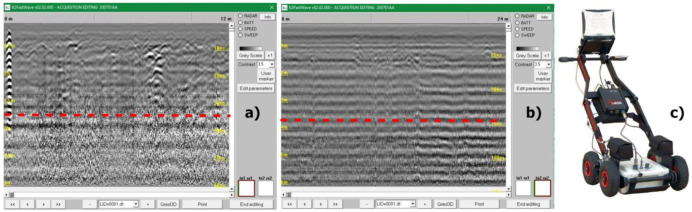
K2 Fastwave sw acquiring data. During crawls along the profiles, reflections caused by subsoil’s discontinuities are represented in real time. The horizontal axis reproduces the antenna advance’s direction, while the vertical axis represents pulse penetration’s direction in nanoseconds. (**a**) Radargram from 600 MHz antenna. (**b**) Same profile recorded with 200 MHz antenna. The difference in terms of resolution and investigation’s depth is noticeable; (**c**) IDS HiMod.

**Figure 6 sensors-23-02769-f006:**
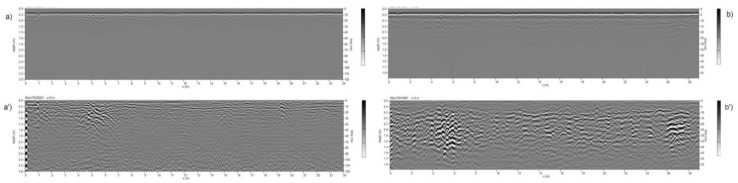
Comparison among radargrams before and after signal processing. (**a**)-(**a’**) 200 MHz; (**b**)-(**b’**) 600 MHz.

**Figure 7 sensors-23-02769-f007:**
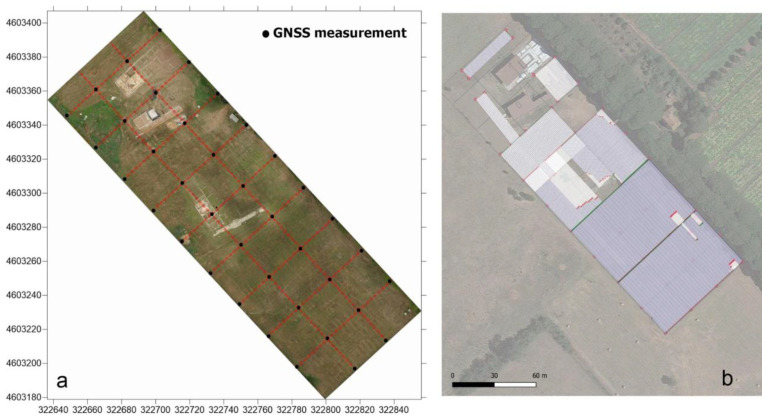
GPR survey planification. (**a**) Area’s subdivision using GNSS in tracing mode. (**b**) Profile orientation and surveyed area.

**Figure 8 sensors-23-02769-f008:**
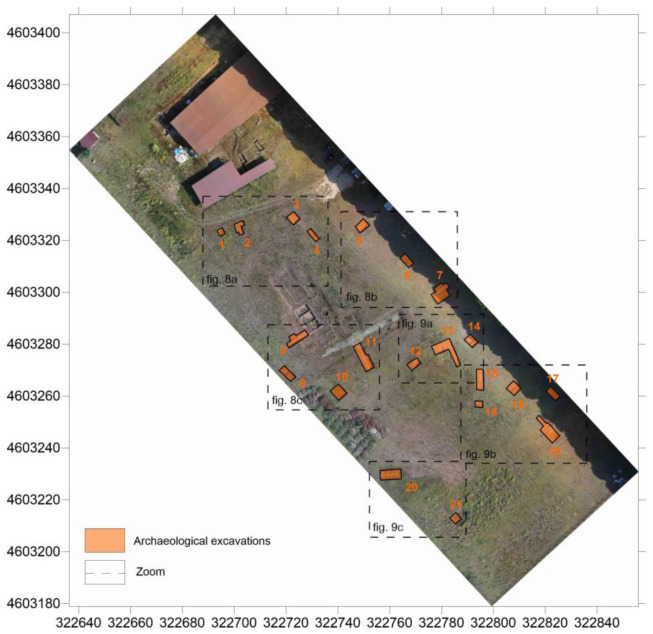
Archaeological excavations position to verify ground truth.

**Figure 9 sensors-23-02769-f009:**
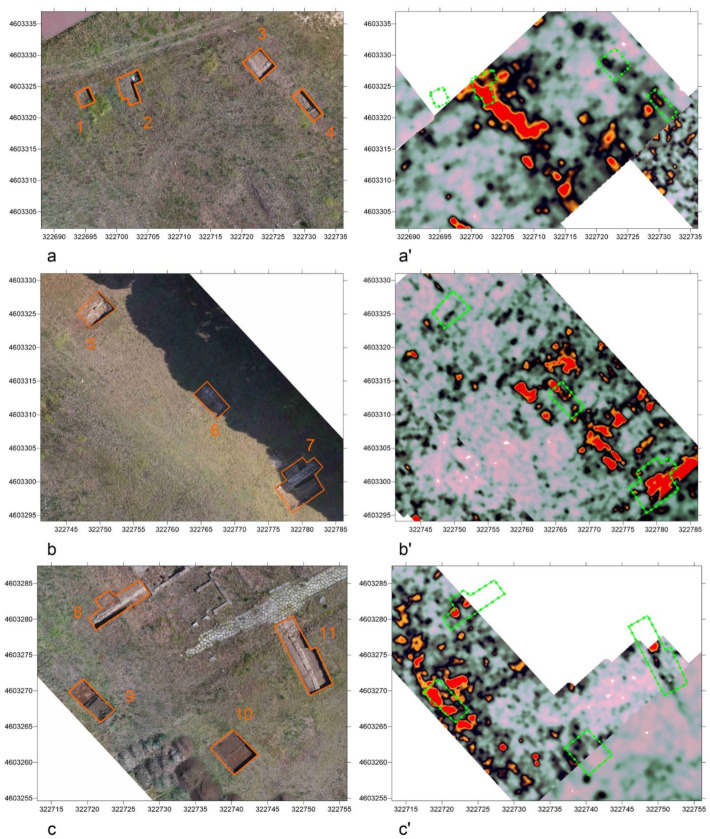
Comparison among ground truth and NW anomalies in the time slices (200 MHz antenna −0.5/−0.7 m). (**a**,**b**,**c**) Ground truth; (**a'**,**b'**,**c'**) GPR anomalies in the same area. Areas in orange and green correspond to areas traced with GNSS prior to excavation.

**Figure 10 sensors-23-02769-f010:**
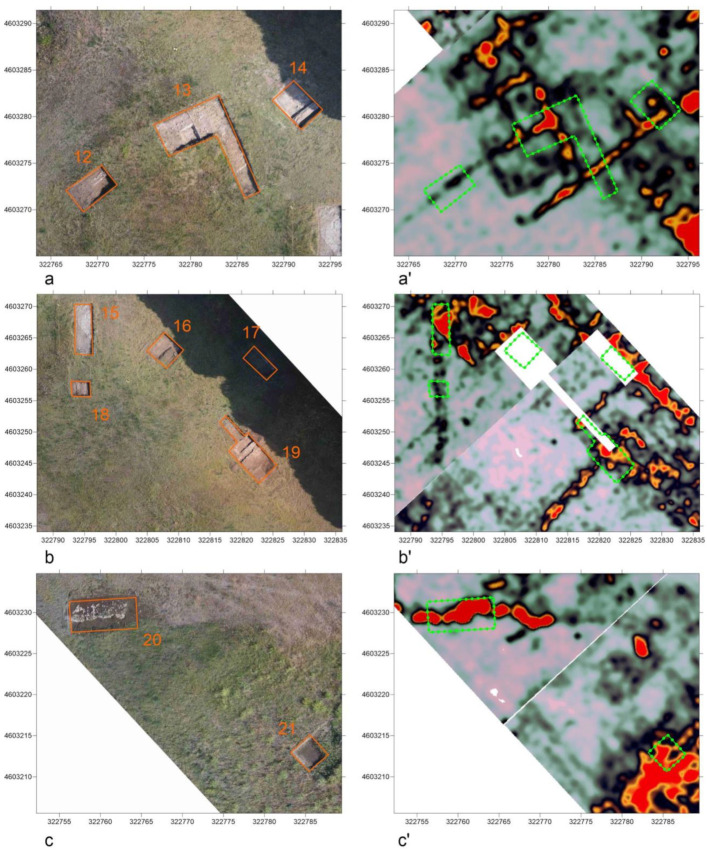
Comparison among ground truth and SE anomalies in the time slices (200 MHz antenna −0.5/−0.7 m). (**a**,**b**,**c**) Ground truth; (**a'**,**b'**,**c'**) GPR anomalies in the same area. Areas in orange and green correspond to areas traced with GNSS prior to excavation.

**Figure 11 sensors-23-02769-f011:**
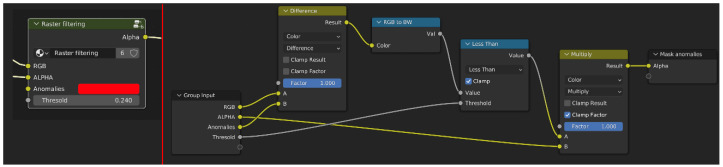
Detail of the data filtering algorithm (node group on the left, exploded algorithm on the right) starting with the input RGB data, a possible prior alpha channel (in this case, the raster clipping, diagonal to the original tiff image), a reference color for the peak value of the anomaly to be highlighted, and a threshold to reduce or increase filtering.

**Figure 12 sensors-23-02769-f012:**
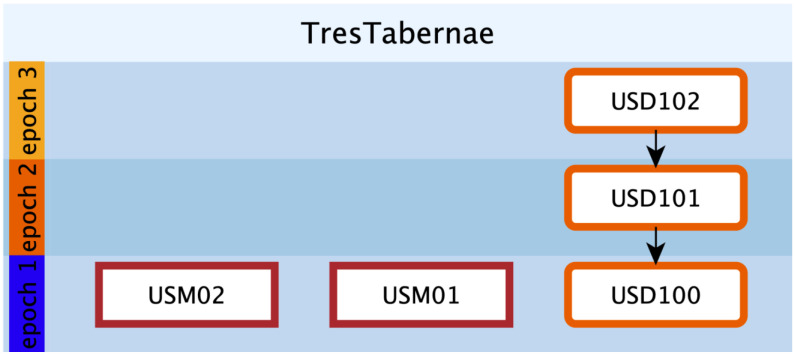
Visualization of EM related to structures found in the subsurface and related supplementary volumes.

**Figure 13 sensors-23-02769-f013:**
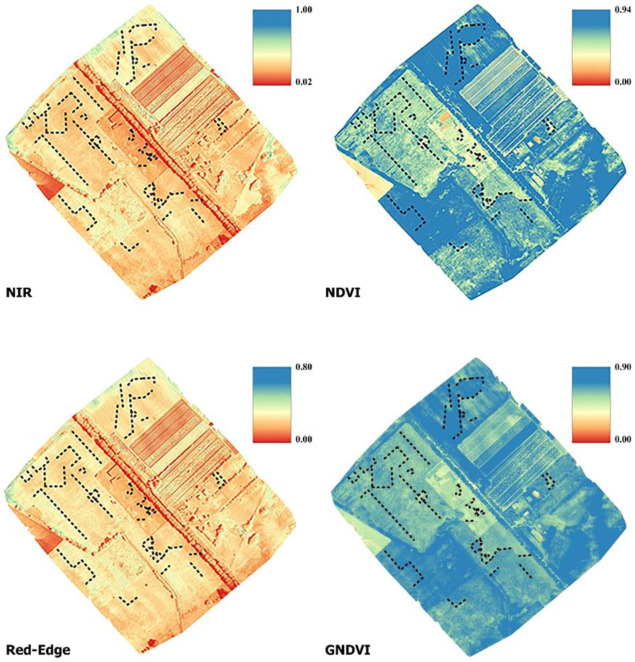
Multispectral maps and detected anomalies (black dotted lines).

**Figure 14 sensors-23-02769-f014:**
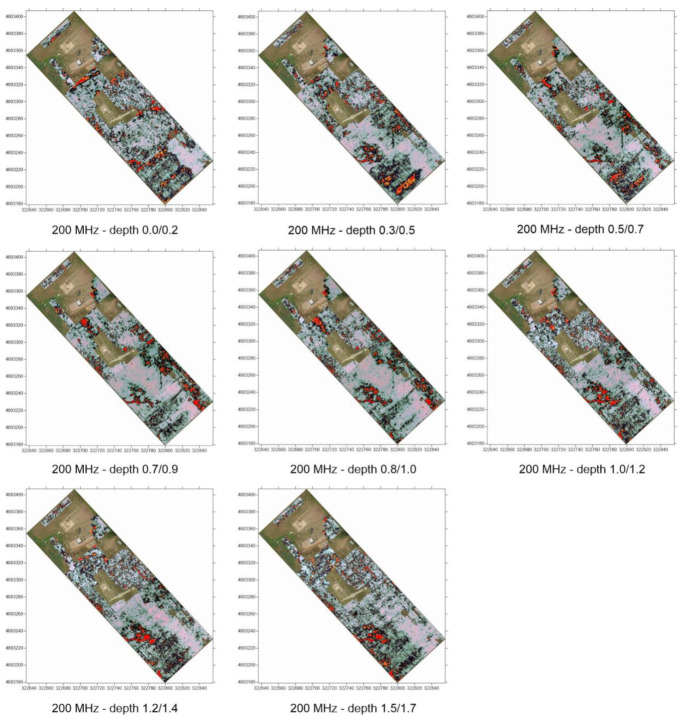
Selection of time slices collected with 200 MHz antenna.

**Figure 15 sensors-23-02769-f015:**
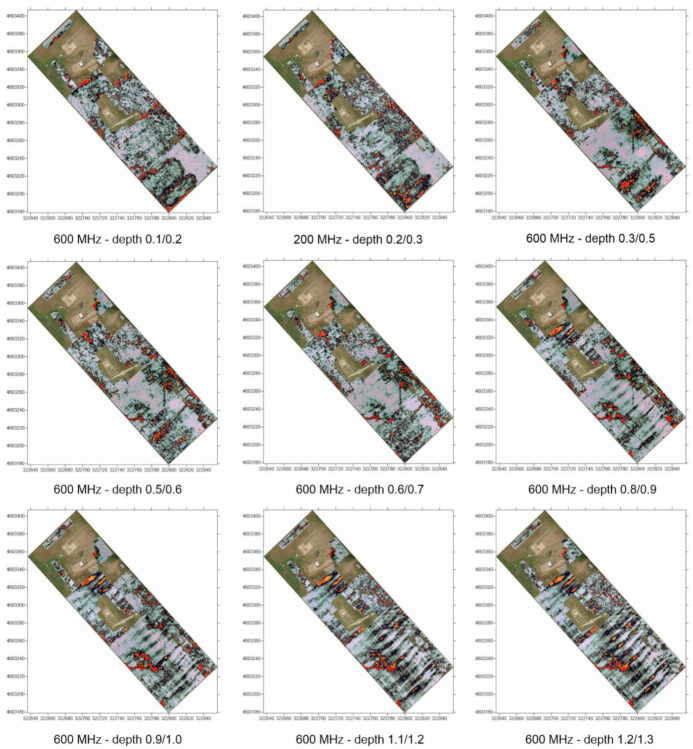
Selection of time slices collected with 600 MHz antenna.

**Figure 16 sensors-23-02769-f016:**
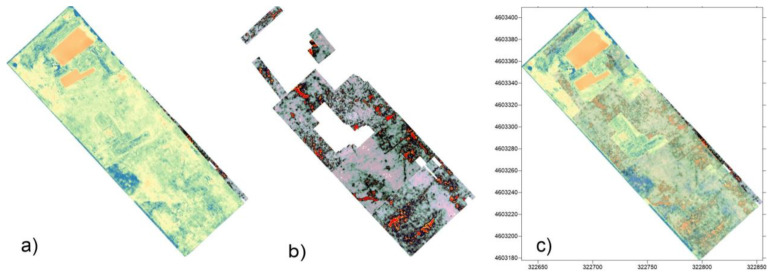
Integration and correspondence of the various information maps produced: (**a**) *NDVI*; (**b**) time-slice 200 MHz (−0.5/−0.7); (**c**) comparison among cropmark anomalies and anomalies identified with GPR.

**Figure 17 sensors-23-02769-f017:**
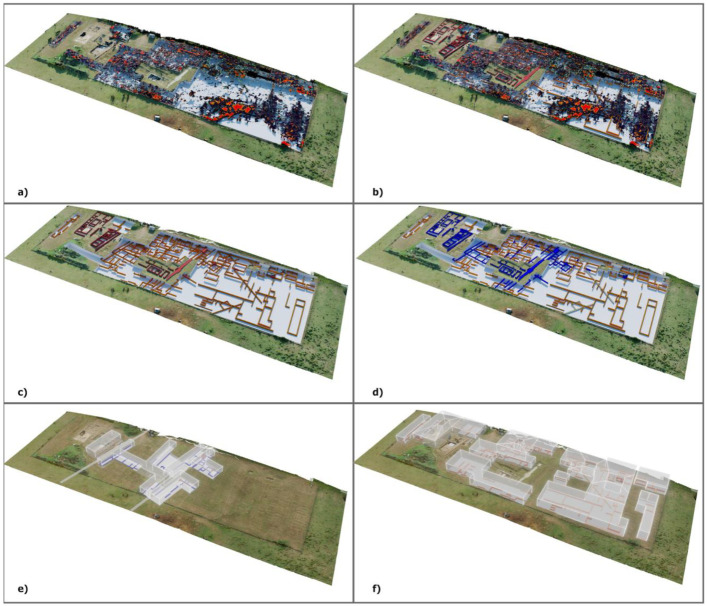
(**a**) GPR time-slices filtered in Blender 3D over sectioned photogrammetric model; (**b**) GPR time-slices filtered in Blender 3D and related proxies over sectioned photogrammetric model; (**c**) proxies over sectioned photogrammetric model. The colors give the different alignments identified and ground truth. (**d**) Interpretation and visualization of the anomalies in the form of proxy volumes (the colors refer to the different chronology). (**e**,**f**) Interpretation and visualization of the anomalies in the form of proxy volumes. The colors refer to the different chronologies.

**Table 1 sensors-23-02769-t001:** Mapping between geophysical anomalies and interpretative lines (direction of the buried walls) and hypotheses by means of volumes (first two columns on the left) with the EM coded formalization by means of colored 3D proxies.

Classical Formal Elements Involved in Geophysics.		EMWgeo Formalization	
2D	3D	Coded Units	Color Code
GPR anomalies	/	USD	Orange
Interpretive lines	Interpretive Volumes	USV/s	Blue
	Volumes depicting potential entire buildings	USV/n	Green

## Data Availability

Not applicable.

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
