# Peer review of "Multispectral UAV Data and GPR Survey for Archeological Anomaly Detection Supporting 3D Reconstruction"

_sensors, 2023, doi:10.3390/s23052769_

Round 1

Reviewer 1 Report

This is a well-presented and really nice example of integrating two methods of data aquisition and interpretation. Some great illustrations. Super material - a way forward for sure.

The grammar is sometimes turtuous: you may lose the reader's attention. It is great material, so why do yourselves the disservice of disguising this in the verbose language? Be succinct: there is a correlation between the high IF journals and high-citation papers and shorter length (everything, title, abstract, text). In this digital age, people do not have the time to navigate long texts with unwieldy language. Shorten and strengthen, for your own benefit.

Some clarity needed between the exposed archaeology and its UAV and/or radar response and what from UAV and/or GPR was defined that is not exposed at surface. I could not see this clearly, and not making it clear opens the authors up for criticism that they are not showing this honestly. OK, I get the comparison message (UAV-RA to geophysics), but the archaeologists reading this will want to know - 'what (new), did you discover?'. If it is simply validation of combing UAV-GPR, then what is the point?

couple of minor questions to consider - check the IDS radar would you? some are one antenna and split the sampling to the two frequencies. just to be sure.

Why discount VARI and other vegetation indices? I see it, but put a statement in, to reassure the reader you are aware, and they exist.

Author Response

We thank the reviewer for the useful suggestions. We truly benefit from your review and have incorporated the suggestions into the text. Below are the responses to each comment.

General comments

This is a well-presented and really nice example of integrating two methods of data acquisition and interpretation. Some great illustrations. Super material - a way forward for sure.

Thank you for the positive comments. The authors are pleased that you enjoyed the paper.

The grammar is sometimes tortuous: you may lose the reader's attention. It is great material, so why do yourselves the disservice of disguising this in the verbose language? Be succinct: there is a correlation between the high IF journals and high-citation papers and shorter length (everything, title, abstract, text). In this digital age, people do not have the time to navigate long texts with unwieldy language. Shorten and strengthen, for your own benefit.

The references have been improved as suggested by the reviewer. The paper was rewritten, and convoluted passages were tightened by shortening the text.

Some clarity needed between the exposed archaeology and its UAV and/or radar response and what from UAV and/or GPR was defined that is not exposed at surface. I could not see this clearly, and not making it clear opens the authors up for criticism that they are not showing this honestly. OK, I get the comparison message (UAV-RA to geophysics), but the archaeologists reading this will want to know - 'what (new), did you discover?'. If it is simply validation of combing UAV-GPR, then what is the point?

The team involved in the research project was not fully involved in this publication. In particular, part of the group involved in the archeological study of the data is in the process of publishing the full results of the research. We did not want to include data that could affect the editorial outcome of their publication. Therefore, this is mainly a methodological paper.

couple of minor questions to consider - check the IDS radar would you? some are one antenna and split the sampling to the two frequencies. just to be sure.

The comments have been taken into account. The references have been improved as suggested by the reviewer.

Why discount VARI and other vegetation indices? I see it, but put a statement in, to reassure the reader you are aware, and they exist.

Other multispectral maps were calculated as LCI-Leaf Chlorophyll Index and NDRE-Normalized Difference Red-Edge, which are less sensitive than the vegetative indices presented and described in the text. For the other vegetative indices such as VARI, there was no way to calculate them because the multispectral camera used did not capture the blue band individually as other indices classified as vegetative RGB indices (TGI-Triangular Greenness Index, GLI-Green Leaf Index, and RGBV-RGB vegetation index for example).

Reviewer 2 Report

I enjoyed reading the manuscript and it is well-structured and written.

Where possible I do suggest improving the quality of some of the figures.

I think the paper can be published in the present form 

Author Response

We thank the reviewer for the useful suggestions. We truly benefit from your review and have incorporated the suggestions into the text. Below are the responses to each comment.

General comments

I enjoyed reading the manuscript and it is well-structured and written.

Thank you for the positive comments. The authors are pleased that you enjoyed the paper.

Where possible I do suggest improving the quality of some of the figures.

According to the reviewer, we found that the resolution of Figure 11 became very low when the text was converted. Figure 11 has been replaced in the text.

I think the paper can be published in the present form 

Thank you for the positive comments.